# Genome-Wide Association Study for Root Morphology and Phosphorus Acquisition Efficiency in Diverse Maize Panels

**DOI:** 10.3390/ijms24076233

**Published:** 2023-03-25

**Authors:** Carlos Alexandre Gomes Ribeiro, Sylvia Morais de Sousa Tinoco, Vander Fillipe de Souza, Barbara França Negri, Christine Marie Gault, Maria Marta Pastina, Jurandir Vieira Magalhaes, Lauro José Moreira Guimarães, Everaldo Gonçalves de Barros, Edward S. Buckler, Claudia Teixeira Guimaraes

**Affiliations:** 1Programa de Pós-Graduação em Genética e Melhoramento, Universidade Federal de Viçosa, Viçosa 36570-000, Minas Gerais, Brazil; 2Embrapa Milho e Sorgo, Sete Lagoas 35701-970, Minas Gerais, Brazil; 3Programa de Pós-Graduação em Bioengenharia, Universidade Federal de São João del-Rei, São João del-Rei 36301-160, Minas Gerais, Brazil; 4Institute for Genomic Diversity, Cornell University, Ithaca, NY 14853, USA; 5USDA-ARS, Robert Holley Center, Ithaca, NY 14853, USA

**Keywords:** GWAS, root morphology, maize, P acquisition efficiency, P deficiency

## Abstract

Maximizing soil exploration through modifications of the root system is a strategy for plants to overcome phosphorus (P) deficiency. Genome-wide association with 561 tropical maize inbred lines from Embrapa and DTMA panels was undertaken for root morphology and P acquisition traits under low- and high-P concentrations, with 353,540 SNPs. P supply modified root morphology traits, biomass and P content in the global maize panel, but root length and root surface area changed differentially in Embrapa and DTMA panels. This suggests that different root plasticity mechanisms exist for maize adaptation to low-P conditions. A total of 87 SNPs were associated to phenotypic traits in both P conditions at −log_10_(*p*-value) ≥ 5, whereas only seven SNPs reached the Bonferroni significance. Among these SNPs, S9_137746077, which is located upstream of the gene GRMZM2G378852 that encodes a MAPKKK protein kinase, was significantly associated with total seedling dry weight, with the same allele increasing root length and root surface area under P deficiency. The C allele of S8_88600375, mapped within GRMZM2G044531 that encodes an AGC kinase, significantly enhanced root length under low P, positively affecting root surface area and seedling weight. The broad genetic diversity evaluated in this panel suggests that candidate genes and favorable alleles could be exploited to improve P efficiency in maize breeding programs of Africa and Latin America.

## 1. Introduction

Food requirements to support continuous population growth are a great challenge for agricultural production. Abiotic stresses are important constraints for global food security, including low phosphorus (P) availability in the soil, which is the second most limiting nutrient for grain yield, after nitrogen [1]. P is a molecular component of nucleic acids, phospholipids, proteins and secondary metabolites, involved with several metabolic processes [2,3], and is therefore essential for plant growth and development. This macronutrient has one of the lowest use efficiency indices in plants mainly due to its high adsorption to soil clays [4]. On tropical soils, which are acid (pH < 5.0) by nature, P is strongly bound to aluminum and iron oxides in the soil minerals, leading to very low P mobility in the soil [5]. Thus, high amounts of P fertilizer are required for agricultural production in high-yield farming systems, where the crops acquire less than 10% of the applied P [6]. Besides the low-P bioavailability to plants, rock phosphate is a non-renewable resource, which may run out in less than 100 years [1,7], justifying studies on P use efficiency and plant adaptation mechanisms to low-P conditions.

Plants developed several strategies to withstand P deficiency [2] that can be classified in two general mechanisms: P internal utilization efficiency and P acquisition efficiency [7,8]. P internal utilization efficiency is related to the unit of yield produced per unit of nutrient applied, which involves mechanisms of P recycling, translocation and storage [3,6,7]. P acquisition efficiency mechanisms are associated with chemical modifications of the rhizosphere, interactions with microorganisms and changes in the root system [9,10,11,12]. The low concentration and mobility of P in the soil make the proliferation and extension of the root system one of the main strategies to maximize P acquisition [13]. Additionally, P acquisition efficiency was more important than P internal efficiency to explain P use efficiency in tropical maize [8], which was confirmed by the co-localization of 80% of the QTLs mapped for P acquisition and P use efficiency in a tropical maize population [14].

Roots perform important functions in plant structure, anchorage and nutrition, showing high diversity in morphology, and dynamic responses to biotic and abiotic stimuli [15,16]. Thus, a promising approach to overcome P deficiency in plants is to explore the diversity in root morphology associated with adaptive mechanisms expressed under low-P conditions. Although useful, root traits are rarely used in plant breeding due to the difficulty in evaluation in field conditions.

Several mapping strategies have been applied to dissect the genetic complexity of root morphology and architecture in maize [17,18,19,20,21,22,23]. Individual roots of young maize plants grown in soil were analyzed and several root type-specific QTLs were identified, that could explain the correlation between root system architecture (RSA) in seedlings and crop yield [17,18,24]. However, a few experiments were performed under P deficiency in nutrient solution [25,26,27,28] and in the soil [29], based on biparental populations, which explore a narrow genetic variability. Among the few genes that effectively promote root development and increase P efficiency, a notable example is the *Phosphorus-starvation tolerance 1* (*PSTOL1*), which encodes a serine/threonine kinase that enhances early root growth and P acquisition in rice [30]. In maize, at least four candidate genes sharing high sequence identity and a conserved serine/threonine kinase domain with *OsPSTOL1* were co-localized with QTLs for root morphology and P acquisition traits [28]. Additionally, other serine/threonine kinases have shown an association with root development [31,32,33,34].

Association mapping emerged as a powerful approach to dissect complex traits, by exploring historical recombination events within a collection of genetically diverse genotypes [35,36]. Maize, which shows impressive genetic diversity, is the most widely cultivated cereal in the world and is highly dependent on P fertilization for sustainable yields, particularly on tropical soils. Thus, the present work aimed to explore root plasticity responses to P in a diverse tropical maize panel and identify candidate genes associated with root morphology and P acquisition traits in nutrient solution.

## 2. Results

### 2.1. Genetic Diversity of the Maize Panels—Principal Component Analysis

The first three principal components based on 12,700 SNPs explained approximately 8.1% of the genetic variance and separated most of the Embrapa lines (black dots in Figure 1) from the DTMA lines (red dots in Figure 1), with some of them clustered together between panels (Figure 1). Maize lines from Embrapa were clustered in three groups, highlighted in dashed black circles in Figure 1, representing the major heterotic groups Flint and Dent, and an intermediate group that shows heterosis with both heterotic groups, named C (Figure 1). These clusters tended to be supported by pedigree information of the maize breeding program (Appendix A). DTMA lines were divided into two groups (dashed red circles in Figure 1), representing maize germplasm from Latin America and Africa, selected based on yield potential and stress tolerance at CIMMYT (International Maize and Wheat Improvement Centre) and IITA (International Institute of Tropical Agriculture) [37]. This population structure may represent the geographical origin and pedigree information of the maize panel, which gather a broad diversity of tropical elite lines currently evaluated in several African and Latin American countries.

### 2.2. Phenotypic Characterization of the Maize Association Panel

A panel composed of 365 maize lines from Embrapa and 196 from DTMA was characterized for three root traits (root length, RL; root surface area, SA, and root diameter, RD), total seedling dry weight (TDW) and total seedling P content (PCont) in nutrient solution under low- and high-P concentrations. TDW and PCont accounted for biomass accumulation and P acquisition efficiency at early stages, respectively, as previously proposed [28,29,38]. All traits had moderate to high broad-sense heritabilities (0.49 to 0.71) (Table 1).

RL positively correlated with SA and both root traits showed positive correlations with TDW under low- and high-P conditions (Table 2). RD positively correlated with SA at both P levels (0.60 and 0.38 for low- and high-P levels, respectively), and negatively with RL in high-P levels as well as with TDW and PCont in both P levels. TDW was highly correlated with PCont, probably because the former is a direct component to calculate PCont. These correlations confirm the strong relationship of the root morphology traits with biomass accumulation and P efficiency in maize seedlings.

### 2.3. Influence of P on Root Morphology and Biomass Accumulation

The genetic divergency between the Embrapa and DTMA panels led us to characterize the phenotypic influence of P separately, for each panel. P treatment significantly influenced all root morphology, biomass accumulation and P efficiency-related traits within the global panel composed of Embrapa and DTMA lines (designated as global panel henceforth) and within the Embrapa lines, whereas only RL and SA were significantly affected by P availability within DTMA lines (Table 3). All root traits increased under high-P levels compared with low-P levels in the global panel (Table 3 and Figure 2). RL and SA were reduced in the Embrapa panel, and increased in the DTMA panel under high-P conditions compared with low-P conditions (Table 3 and Figure 2), showing a differential modification in the root system modulated by P availability (Table 3 and Figure 2). As the genetic variance for these root traits was much higher in the DTMA panel than in the Embrapa panel (Table 3), the positive effect of P for these traits in the DTMA reflected the positive effect in the global panel. RD increased under high-P concentrations compared with low-P concentrations within the Embrapa and global panels, without a significant change within the DTMA panel (Table 3 and Figure 2).

TDW and PCont increased in high-P compared with P deficiency in the global and Embrapa panels, without significant change in the DTMA panel (Table 3 and Figure 3). For these traits, the Embrapa lines presented higher genetic variance compared with the DTMA panel (Table 3), reflecting a wider dispersion of phenotypic values in the boxplot (Figure 3).

### 2.4. Multiple Testing and Type-I Error Corrections for GWAS

Linkage disequilibrium (LD) decayed to basal levels (*r*^2^ = 0.2) in approximately 1000 bp (Figure 4), following a similar pattern for the global, Embrapa and DTMA panels (Appendix A). The LD decayed uniformly among chromosomes, ranging from *r*^2^ = 0.182 on chromosome 7 to *r*^2^ = 0.236 on chromosome 5 based on SNPs separated by 1000 bp. The average LD decay of 1000 bp resulted in 58,708 LD blocks along the genome, which was considered as the number of independent tests. These LD blocks generated a –log_10_(*p*-value) threshold of 6.07 based on the Bonferroni correction.

The first five principal components (PC) explained 10.7% of the total genetic variance, and after the fifth component, the percentage of variance explained was below 1% for each PC (Appendix A). Furthermore, the principal component analysis clustered the maize lines into five major groups, three composed of Embrapa lines and two of DTMA lines (Figure 1), which were consistent with the geographical origin and pedigree information. Thus, the first five PCs were used as fixed covariates in the GWAS analyses to account for population structure (Q). The model selected for type-I error correction was the mixed linear model (MLM) including population structure, based on the first five principal components (PC), and familial relatedness (K), which presented the best fit in the QQ-plot for RL in low-P (Appendix A). This profile was similar for all traits and conditions, indicating that the PC + K model was effective to control false positives associations for all combinations of traits and P conditions.

### 2.5. SNPs Significantly Associated with Root Morphology, P Acquisition and Biomass Related-Traits

Out of 353,540 SNPs tested, 87 were associated to all five phenotypic traits in both P conditions at—log_10_(*p*-value) ≥ 5 (Appendix A), and seven SNPs reached the Bonferroni significance threshold of −log_10_(*p*-value) ≥ 6.07 (Table 4 and Appendix A).

The SNP S4_3751192 was significantly associated with RD under high-P levels [−log_10_(*p*-value) = 6.44], and was placed within the predicted gene GRMZM2G037472 that encodes a kinesin motor domain containing protein, putatively involved in microtubule motor. The SNP S8_88600375 was associated with RL under low-P [–log_10_(*p*-value) = 6.30] and was located within the predicted gene GRMZM2G044531, which encodes an AGC kinase family protein.

For TDW, two SNPs were associated under low-P conditions, S6_34607369 and S9_137746077, whereas the S10_77284783 was associated under high-P condition. The SNP S9_137746077 [−log_10_(*p*-value) = 6.43] was located 57 bp upstream the gene GRMZM2G378852 that encodes a putative MAPKKK family protein kinase. The SNP S10_77284783 was mapped within the predicted gene GRMZM2G110145, a putative cellulose synthase-like family protein. The change in the cellulose synthase pathway was one of the mechanisms for low-P tolerance in maize revealed by a combination of GWAS and transcriptome analyses [39].

Two SNPs were associated with PCont under high-P conditions, S9_143192439 located within the uncharacterized predicted gene GRMZM2G104618, and S8_21326857 mapped at bin 8.03 without flanking genes within the target genomic region.

### 2.6. Exploring the Genomic Region Flanking the SNP S8_8600375

The genomic region of approximately 600 kb on maize chromosome 8 (bin 8.03) encompassing the SNP S8_88600375, which was highly associated with RL under low-P levels and presented a consistent peak in the Manhattan plot (Appendix A), is shown in Figure 5. This region spanned 30 SNPs covering three predicted genes (GRMZM2G057116, GRMZM2G044531 and GRMZM2G104133) with a low LD (*r*^2^ = 0.18 in 1000 bp), similar to the overall mean found along the genome (*r*^2^ = 0.2). Interestingly, the SNP S8_88600375 (GRMZM2G044531 encoding a putative AGC kinase) presented high LD (*r*^2^~0.65) with seven SNPs located within the predicted gene GRMZM2G057116, which encodes a WRKY transcription factor, located more than 120 kb away from it (Figure 5). One of these SNPs, S8_88472064, was associated with RL under low-P conditions, within the peak but below the threshold [–log_10_(*p*-value) = 4.68], and shared a high LD (*r*^2^ = 0.77, D′ = 0.90) with S8_88600375.

## 3. Discussion

Maize is a cross-pollinated species that normally presents fast LD decay [40] that is highly dependent on the genetic diversity of the panel. The LD decay in our maize panel was on average 1 kb considering an *r*^2^ threshold of approximately 0.2, which was similar to the extent of LD obtained in the entire collection of maize lines conserved at the USDA [41] and to the DTMA-AM panel composed by 278 diverse maize lines [42]. A large variation of LD decay was found depending on the germplasm, which was slower within the stiff stalk germplasm (10 kb for *r*^2^~0.2) and faster within tropical materials, reaching 1 kb for the same *r*^2^ [41]. The LD decay was similar for both maize panels from Embrapa and DTMA, but the genetic variance for root traits was superior in DTMA panel whereas the genetic variance for total seedling dry weight and P content was superior in the Embrapa panel.

### 3.1. Root and Biomass Plasticity under P Deficiency

The root traits (root length RL, root surface area SA and root diameter RD) were characterized in nutrient solution with images analyzed using RootReader2D and WinRHIZO software, and the seedlings were measured for total seedling dry weight (TDW) and P content (PCont). Interestingly, the range of phenotypic values for RL (20.8–515.5 cm) and TDW (15.3–220.3 mg) were very similar to the ones obtained for 384 temperate maize lines belonging to Ames panel (RL: 16.4–536.3 cm and TDW: 16–253 mg), whose root systems were evaluated with the *ARIA—Automatic Root Image Analyzer* [20]. The similarity of phenotypes between two highly diverse panels characterized with different methods indicates good consistency and reliability of the root morphology measurements in the paper pouch system under hydroponics as well as of the biomass accumulation at the seedling stage.

Maize lines from Embrapa significantly modified the root morphology under P deficiency that showed longer (increased RL and SA) and finer roots (reduced RD) compared with the high-P conditions. These changes in root morphology are expected to improve P acquisition efficiency due to the low mobility of phosphates in the soil, which could enhance the adaptation to low-P soils, prevalent in the tropical regions of Brazil. The ability to overcome low-P availability has been associated with the exploratory capacity of roots in the soil [8,9,11,12,13,43]. This relationship was emphasized in tropical maize lines, where the P acquisition efficiency was more relevant than P internal utilization efficiency to explain the total variation in P use efficiency under low-P soils [8]. The root system plays a major role in crop performance and the distribution of roots close to the nutrient-rich zone favors the plant establishment. Therefore, root morphology, architecture and plasticity become very important for water and nutrient uptake [4].

Nonetheless, maize lines from DTMA showed reduced RL and SA under P deficiency, which could be explained by other mechanisms, considering a much larger genetic variance compared with the Embrapa panel. Different plasticity in response to varying soil P concentrations indicates diversification of P acquisition strategies across species [44]. Adaptations in root morphology, P-mobilizing exudates and mycorrhizal symbioses contribute to enhance plant P absorption, but in contrasting ways, presumably with different costs and benefits [4,45,46]. The spatial arrangements of root system architecture can also influence the microbiota and play a critical role in plant productivity and tolerance to environmental stresses [4,47]. Additionally, root morphology-related genes were selected during maize domestication and breeding, leading to the accumulation of favorable alleles to develop a root system suitable for high-density planting [21]. Thus, the genetic diversity of both panels combined with different origin and breeding sources is probably related to the differential root plasticity under contrasting P availability between the Embrapa and DTMA panels.

In the global and Embrapa panels, TDW and PCont was enhanced under high-P compared with low-P treatment, probably due to the higher P availability in the nutrient solution for biomass accumulation at the seedling stage. The superior biomass accumulation and P content under high-P availability was consistent with the positive correlation between these traits under both P treatments. The large genetic variance in the Embrapa panel for these traits contributed to the significant effect of P in the global panel.

### 3.2. Candidate Genes Associated with Root Morphology and P Acquisition-Related Traits

Diameter is an important feature of root systems in plants. The SNP S4_3751192 that reduced RD under high-P conditions tended to improve TDW and PCont (Appendix A), in agreement with the negative correlation of RD with both seedling biomass and P content. The genomic region of this SNP could be considered coincident with the SNP associated with a 100-grain weight (chromosome 4 at 3.6 Mb) in a panel composed by 410 maize lines cultivated under normal P conditions [39]. Thus, the overall contribution of root diameter could result in positive gains for maize yield under different environment conditions. A small diameter of seminal roots confers advantage for nutrient acquisition [11], whereas a large diameter of primary root facilitates water flux and root penetration in hard soils, improving drought tolerance and nitrogen acquisition [48,49]. Additionally, finer roots interact with and modify the surrounding soil environment through the exudation of labile C from living roots, which stimulate microbial activity and mediate the dynamics of short- and long-term pools of soil organic C [50].

Increases in root length and root surface area enhanced P acquisition in *Brassica napus* [51], rice [30], sorghum [52,53,54], maize [28] and barley [55]. The C allele of SNP S8_88600375 significantly enhanced RL under P deficiency, also positively affected SA and TDW, without affecting RD (Appendix A). This SNP was mapped within GRMZM2G044531, which encodes a putative AGC kinase family protein. Different kinases from AGC family have been described in Arabidopsis to modulate root hair development [31], to promote root growth in response to mycorrhiza fungal infection [32] and to regulate root waving [34]. This SNP co-localizes with a QTL (RL8-KXDP) for taproot length in a Chinese F_2:3_ population cultivated under P deficiency [38].

The A allele of the SNP S9_137746077 significantly improved TDW, and contributed positively with RL and SA under P deficiency (Appendix A). This SNP was located 57 bp upstream of the maize gene GRMZM2G378852 that encodes a putative MAPKKK family protein kinase. MAPKKK is a component of mitogen-activated protein kinase cascade involved on signal transduction to translate external stimuli into cellular responses. Several MAPKKKs were involved in signaling pathways in different maize organs and developmental stages [56]. Genome-wide studies revealed MAPKKK genes expressed in primary and crown roots associated in drought responses in a Chinese maize line highly tolerant to drought stress [57]. This SNP encompasses a genomic region associated with grain number in a maize panel cultivated under normal P supply in two years [39].

The SNPs S9_137746077 and S8_88600375 reflected the positive correlations of root morphology traits (RL and SA) with biomass accumulation (TDW) under P deficiency (Table 2). These positive correlations were also observed in maize RILs cultivated under P deficiency [28] and in the Ames maize panel evaluated under sufficient P supply [20]. In maize, the elongation of the main root axis is maintained under P starvation [58], allowing more P acquisition and biomass accumulation [9]. These SNPs encompass two putative serine/threonine kinases (MAPKKK kinase, GRMZM2G378852, tagged by S9_137746077 and AGC kinase, GRMZM2G044531, tagged by S8_88600375), which share a similar kinase domain with PSTOL1 (Phosphorus Starvation Tolerance 1), a key protein to enhance early root growth and P acquisition in rice [30] and sorghum [52]. Additionally, other protein kinases have been described to be directly or indirectly involved in pathways related to root development [32,33,59], suggesting a relevant role of this protein family on regulating root morphology.

### 3.3. Exploring SNPs at Bin 8.03

The genomic region on maize chromosome 8 (bin 8.03) flanking the SNP S8_88600375 (GRMZM2G044531 for AGC kinase), which was highly associated with RL under low-P levels, presented a low LD (*r*^2^ = 0.18 in 1 kb), similarly to the overall mean of LD decay along the genome (*r*^2^ = 0.2). However, this SNP is in high LD (*r*^2^~0.77) with three SNPs within the predicted gene GRMZM2G057116 (WRKY transcription factor) that were separated by ~127 kb of S8_88600375 (Figure 5). Although high variation in LD is expected for breeding lines, the high LD between both predicted genes, AGC kinase and WRKY transcription factor, may suggest a hypothetical relationship between them. WRKY transcription factors have been involved in the regulation of several developmental processes and responses to biotic and abiotic stresses in plants [60,61,62,63,64]. The WRKY family was characterized in maize, in which the candidate gene GRMZM2G057116 was phylogenetically clustered at Group IIc and was expressed at low levels in several maize tissues [65]. This candidate gene was also downregulated under cycles of dehydration–rehydration in 14-day-old maize seedlings [66]. Additionally, one W-box element (cTGACc) was located at 1773 bp upstream the predicted start-codon of this AGC kinase, indicating that the AGC kinase can be regulated by the WRKY transcription factor. However, functional studies are required to validate this hypothesis.

SNPs associated with root morphology, seedling biomass and P content were discussed in the light of candidate genes and genomic regions previously associated with such traits. As root morphology and P acquisition efficiency are not easy traits to evaluate under field conditions, strategies to investigate the genetic basis underlying these traits in young plants could help the identification of genes and favorable alleles useful for breeding purpose to enhance yield performance under P deficiency.

## 4. Materials and Methods

### 4.1. Plant Material

The maize association panel composed of 561 maize inbred lines was designated as the global panel. This global panel was composed of 365 lines from the Embrapa maize and sorghum and 196 lines from the drought-tolerant maize for Africa panel (DTMA) project. The maize lines from Embrapa included advanced breeding lines and founder parents of populations contrasting for yield performance under low P [14,67] as well as six lines from CIMMYT (CML), two temperate lines and one Cateto line from Colombia. The other group was a random subset of DTMA represented maize lines from CIMMYT (International Maize and Wheat Improvement Centre) and IITA (International Institute of Tropical Agriculture) [37]. A complete list of the maize lines including their pedigree and origin is shown in Appendix A.

### 4.2. Root Morphology Analysis, Phosphorus and Biomass Quantification

Root morphology traits in the panel were assessed using a paper pouch system in hydroponics as described in [68]. Maize seeds were surface sterilized with sodium hypochlorite 0.5% (*v*/*v*) for 5 min and germinated in moistened paper rolls. After four days, the endosperm was removed and seedlings were transferred to moistened blotting paper placed in paper (24 × 33 × 0.020 cm). Each container accommodated 10 paper pouches with three uniform seedlings per pouch, whose bottoms were immersed in a half strength Magnavaca’s nutrient solution [69] with 2.5 μM and 250 μM P for low- and high-P treatments, respectively. The nutrient solution was replaced every three days and the pH was maintained at 5.65. The containers were maintained under continuous aeration in a growth chamber with 27/20 °C day/night temperatures, 12 h of light and dark and light intensity of 330 µmol photons m^−2^ s^−1^.

After 13 days, the root images were captured with a digital photography setup and analyzed using RootReader2D ([70], http://www.plantmineralnutrition.net/rootreader.htm, accessed on 17 November 2014) and WinRHIZO (http://www.regent.qc.ca/, accessed on 17 November 2014) software. The imaging system consists of a Nikon (Melville, NY, USA) D300s digital SLR Camera with 60 mm macro lens, calibrated for a fixed focal plane scale of 140 pixels cm^–1^. The root systems of each plant were spread out in a clear, water-filled tray that was illuminated from below and individually imaged. The captured images were batch-converted to an eight-bit grayscale format using a custom written ImageJ (http://www.plantmineralnutrition.net/, accessed on 17 November 2014) plugin. The images were then batch-thresholded using the RootReader2D software and the images imported into the WinRHIZO software for root trait analysis and quantification. The images were analyzed using a calibration grid as a reference scale and changing the input settings to pale roots on a black background.

Out of 24 different root traits generated by the software, only three were used in our study. The total root system, which comprises primary, seminal and initial adventitious roots, was evaluated for total root length (RL, cm), total root surface area (SA, cm^2^) and average root diameter (RD, mm). The final measurements were the average of three replicates. These root traits were selected due to their relevance to represent the root system at seedling stages under P deficiency, as previously discussed in maize [28,68], and importance for associations to P efficiency in sorghum [52,53].

Root and shoot tissues were dried separately at 65 °C in a forced-air oven until constant weight, then joined to determine the total seedling dry weight (TDW). For the P concentration analysis, plant tissues were digested by nitric perchloric acid and quantified using inductively-coupled argon plasma (ICP) emission spectrometry. Total phosphorus content (PCont) was calculated by multiplying total seedling dry weight and P concentration.

### 4.3. Experimental Design and Adjusted Means

The maize lines were phenotyped in three different experiments, containing 192, 181 and 212 genotypes, with 12 common checks. The experimental design was in groups of experiments, with each experiment arranged in randomized complete blocks with three replicates, under low- and high-P conditions.

Adjusted means of inbred lines were obtained via Best Linear Unbiased Estimates (BLUE) for each trait and P level were used to obtain the adjusted means using GenStat 16.1 [71] according to Model 1:(1)$$y_{ijk}=\mu +G_{i\left(j\right)}+E_{j}+R_{k\left(j\right)}+\varepsilon_{ijk}$$$y_{ijk}$ is the phenotype of the $i^{th}$ line (*i* = 1…561), in the $j^{th}$ experiment (*j* = 1, 2, 3), of replicate *k* (*k* = 1, 2, 3); μ is the overall mean; $G_{i\left(j\right)}$ is the fixed genotypic effect of individual *i* at experiment *j*, including the effects of lines and common checks; E*_j_* is the fixed effect of experiments; $R_{k\left(j\right)}$ is the fixed effect of replicate *k* at experiment *j*; and $\varepsilon_{ijk}$ is the random non-genetic effect ε*_ijk_*~*N*(*0,* $\sigma_{\varepsilon }^{2}$).

Different structures of variance–covariance (VCOV) matrices were compared for the residual effect and the VCOV model with heterogeneous variance and uniform covariance among experiments was selected based on AIC (Akaike Information Criterion) [72]. Diagnostic plots were used to verify whether the residuals of the fitted models were normally distributed and the presence of outliers.

The broad-sense heritability (Table 1) and Pearson correlation (Table 2) were calculated using GenStat 16.1 [71]. The broad-sense heritability was calculated according to Expression (2):(2)$$h^{2}=\sigma_{g}^{2}/(\sigma_{g}^{2}+\sigma_{\varepsilon }^{2}/r)$$
where $\sigma_{g}^{2}$ is the genetic variance component, $\sigma_{\varepsilon }^{2}$ is the error variance component and r the number of replicates. The variance components were obtained through the linear model described above, except that the genotypic effect was considered as random.

Additionally, joint analyses of variance were performed to test the effects of P levels within each panel (Embrapa and DTMA) and within the global panel. Phenotypic variability within panel and overall were examined through boxplots. All these phenotypic analyses were performed using GenStat 16.1 [71].

### 4.4. SNP Data

Genomic DNA was isolated from young leaves using the CTAB method [73] and submitted to genotyping-by-sequencing (GBS) using the *Ape*KI restriction enzyme for library preparation [74] at the Genomic Diversity Facility, Cornell University. The raw sequences were processed using the TagsOnPhysicalMap files (TOPM) in the GBS production pipeline as proposed by [75], based on the maize B73 RefGen_v3 genome sequence.

For association mapping, missing data in the 955,120 SNPs were imputed using the Fast Inbred Line Library Imputation (FILLIN) method in TASSEL version 5.2.21 [76], with the default parameters. The imputed data were filtered for minor allele frequency (MAF) higher than 5%, resulting in 353,540 SNPs. A schematic flowchart of the SNP filtering process is presented in Appendix A.

### 4.5. Population Structure and Familial Relatedness

Population structure and familial relatedness analyses were carried out with 12,700 randomly selected non-imputed SNPs, filtered for less than 10% of missing data and for MAF ≥ 1%. Principal component analysis was used to study population structure in the maize association panel using the software TASSEL version 5.2.21 [76]. The first three principal components obtained in the principal component analysis (PCA) were plotted in a 3D graphic for clustering analysis and the first five principal components (PC) were used to control population structure in the association analysis. Familial relatedness or kinship (K) matrix was calculated using a pairwise dissimilarity matrix based on the simple matching coefficient in TASSEL version 5.2.21 [76], which was converted to a similarity matrix by subtracting all values from 2 and then scaling so that the minimum value in the matrix is 0 and the maximum value is 2 [35].

### 4.6. Linkage Disequilibrium (LD)

LD decay ($r^{2}$) was calculated for non-imputed 29,188 SNPs filtered for MAF ≥ 5% and missing data ≤ 10% using TASSEL version 5.2.21 [76]. Nonlinear models of *r*^2^ values for SNPs separated by pairwise distances ranging from 1 bp to 1 Mb were calculated and displayed in a violin plot per chromosome and for all 10 chromosomes. The resulting models were depicted using the package *vioplot* [77] available in the R software [78], considering seven distance intervals (DI) of 1 bp; 1 < DI ≤ 10 bp; 10 < DI ≤ 100 bp; 100 < DI ≤ 1000 bp; 1000 bp < DI ≤ 10,000 bp; 10,000 < DI ≤ 100,000 bp and 100,000 < DI ≤ 1,000,000 bp. SNPs with $r^{2}$ < 0.2 were considered in linkage equilibrium.

### 4.7. GWAS Analysis

Four linear models were tested to control for false positive associations (type-I error). The naïve model is a general linear model (GLM) with no correction for population structure or familial relatedness: y=Xβ+e*;* the GLM model incorporating population structure (PC) is y=Xβ+Qv+e; the mixed linear model (MLM) accounting for familial relatedness (K) is y=Xβ+Zu+e and the MLM incorporating both PC and K (PC + K) is y=Xβ+Qv+Zu+e. In these models, *y* is a vector of phenotypic values, *β* is a vector of the fixed effects of markers, *e* is a vector of residual effects, *v* is a vector of fixed effects related to population structure and *u* is a vector of polygene background random effects related to familial relatedness. *X* and *Z* are the incidence matrices of 0 s and 1 s, relating to *β* and *u*, respectively, to *y*. Q is a matrix of population structure matrix obtained by the principal component analysis and K is the kinship (co)variance matrix between pairs of inbred lines [35,76]. Model selection and the association analyses were performed using TASSEL version 5.2.21 [76]. Model fitness was based on visual inspection of quantile-quantile plots (QQ-plot) of observed versus expected −log_10_(*p*-value) distributions.

The Bonferroni procedure was used for multiple testing correction, considering a significance level (α) of 0.05 and the number of independent tests, which were estimated based on the number of independent LD blocks along the maize chromosomes according to [79]. Candidate genes were searched considering a local LD estimated for each SNP using the maize B73 RefGen_v3 genome sequence.

## 5. Conclusions

The contrasting levels of P modified root morphology traits, biomass and P content in the global maize panel, but root length and root surface area changed differentially within Embrapa and DTMA panels, suggesting different root plasticity mechanisms for adaptation to low-P conditions. Out of 353,540 GBS-based SNPs, only seven reached the Bonferroni significance threshold, including two SNPs significantly associated with total seedling dry weight and with root length under P deficiency, which tagged a MAPKKK and an AGC protein kinases. These associations enforce the importance of protein kinases in root development changes modulated by P deficiency, such as those found for PSTOL1, a protein kinase that enhances root growth and P acquisition in rice and sorghum. The diversity panel composed by maize elite lines from Embrapa and DTMA suggest that elite lines and favorable alleles could be exploited to improve P efficiency in maize breeding programs of Africa and Latin America.

## Figures and Tables

**Figure 1 ijms-24-06233-f001:**
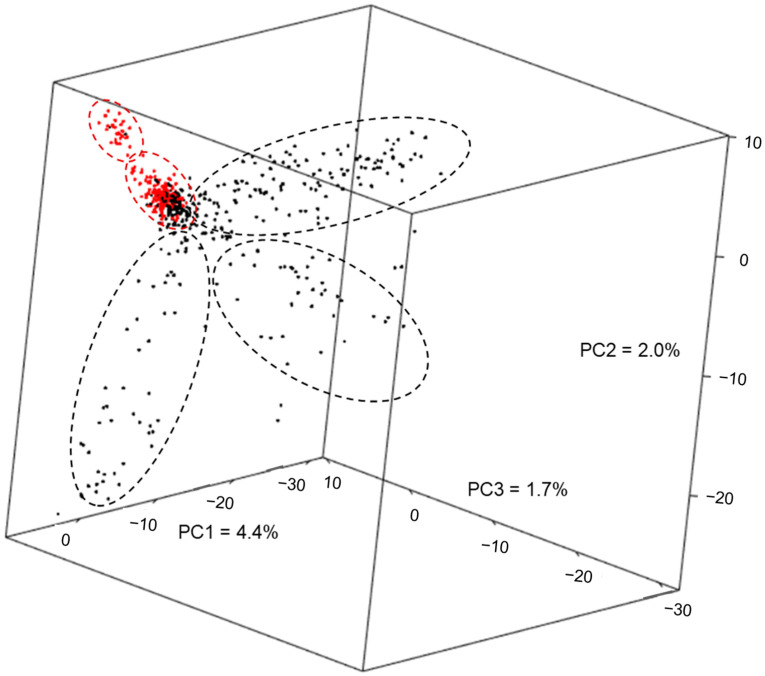
Principal component analysis of 563 inbred lines, including 365 lines from Embrapa (black dots) and 198 from DTMA panel (red dots) based on the first three principal components (PC). The percentages represent the genetic variance explained by each PC. The dashed black circles indicate three clusters of Embrapa lines (black dots) representing Flint, Dent and C groups; and the dashed red circles indicate two clusters of DTMA lines (red dots).

**Figure 2 ijms-24-06233-f002:**
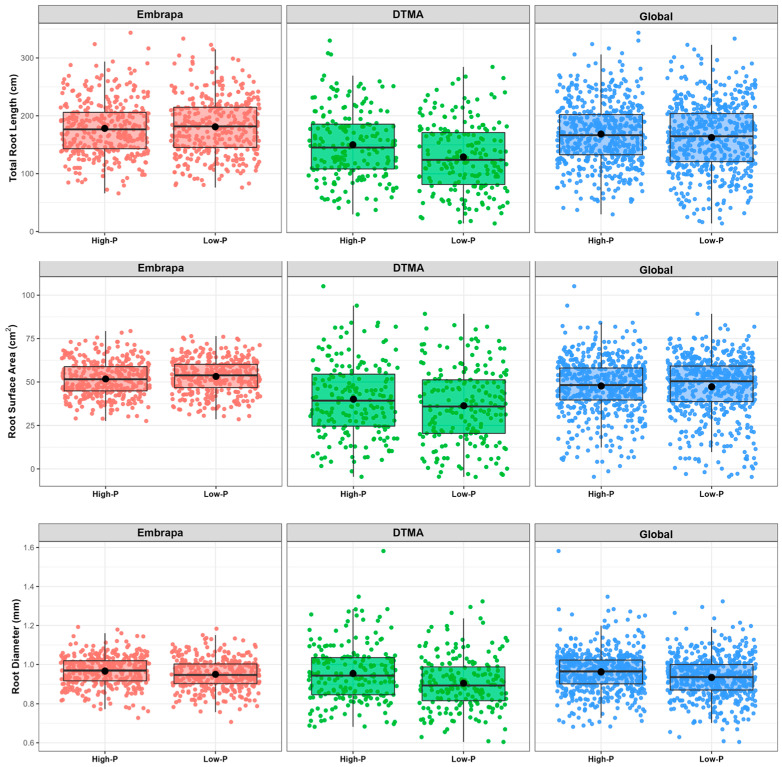
Boxplots showing the distribution of root morphology traits in the Embrapa, DTMA and for the global maize panels in high- and low- P concentrations. Phenotypes are BLUEs for root length in cm (RL), root surface area in cm^2^ (SA) and root diameter in mm (RD).

**Figure 3 ijms-24-06233-f003:**
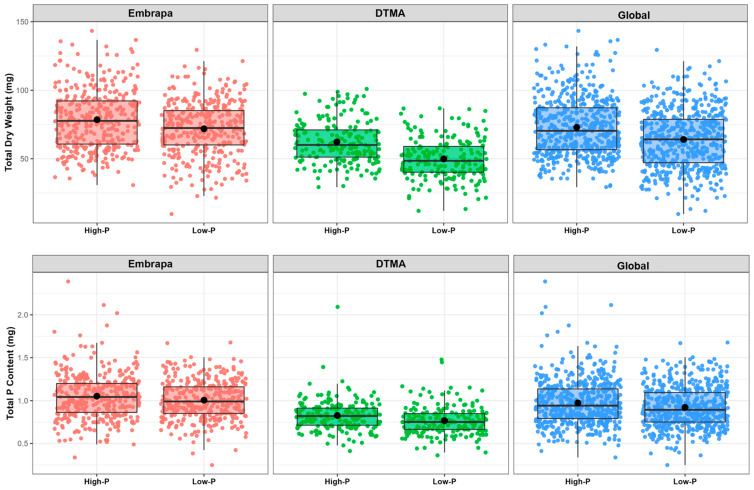
Boxplot showing the biomass accumulation and P content in the seedlings distribution over the Embrapa, DTMA and global maize panels in high- and low-P concentrations. Phenotypes were BLUEs for total seedling dry weight in mg (TDW) and total P content in mg (PCont).

**Figure 4 ijms-24-06233-f004:**
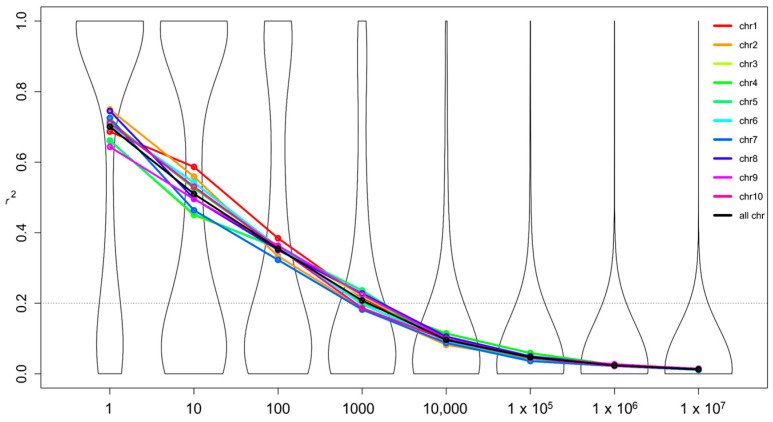
Linkage disequilibrium decay across the 10 maize chromosomes based on 29,188 SNPs. The black line represents the average of all chromosomes, and the violin plot showing the distribution of all *r*^2^ values is depicted in the background. Dotted lines represent the threshold of 0.2.

**Figure 5 ijms-24-06233-f005:**
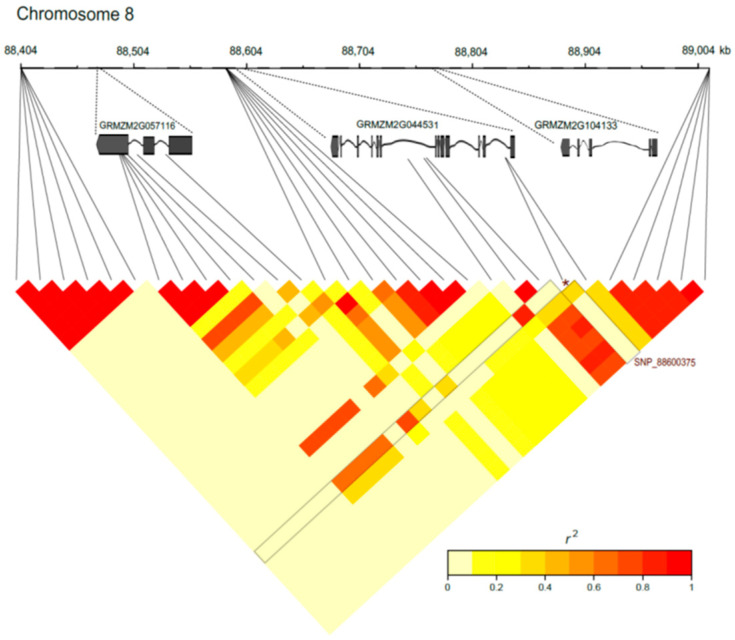
Linkage disequilibrium in a 609.6 kb region within the bin 8.03 of the chromosome 8. The black lines represent each SNP and the asterisks depict the most significant SNP of this region, SNP S8_88600375, for root length under low-P conditions. The positions of the predicted genes GRMZM2G057116 (WRKY member), GRMZM2G044531 (AGC kinase) and GRMZM2G104133 are shown.

**Table 1 ijms-24-06233-t001:** Phenotypic mean, minimum (Min) and maximum (Max) values, genetic ($\sigma_{g}^{2}$ ) and residual ($\sigma_{\varepsilon }^{2}$ ) variances, coefficient of variation (CV, %) and broad-sense heritability ($h^{2}$ ) of the five traits evaluated under low- and high-phosphorus (P) levels in 561 tropical maize lines.

Trait	Acronym	P Level	Mean	Min	Max	$\sigma_{g}^{2}$	$\sigma_{\varepsilon }^{2}$	CV	*h* ^2^
Root Length (cm)	RL	Low	162.5	32.4	479.4	1807	1935	39.7	0.71
		High	168.4	20.8	515.5	1858	2017	44.7	0.71
Root Surface Area (cm^2^)	SA	Low	47.2	12.0	200.8	60.2	175.2	55.0	0.54
		High	47.7	9.2	154.0	93.9	165.3	48.4	0.65
Root Diameter (mm)	RD	Low	0.93	0.37	2.15	0.005	0.810	33.4	0.70
		High	0.96	0.43	2.25	0.006	0.704	32.5	0.70
Total Seedling Dry Weight (mg)	TDW	Low	64.1	11.5	166.3	198.0	227.6	36.3	0.70
		High	72.9	15.3	220.3	263.4	327.6	40.9	0.70
Total P Content (mg)	PCont	Low	0.92	0.10	1.32	0.02	0.017	36.4	0.49
		High	0.97	0.07	2.39	0.03	0.019	41.1	0.56

**Table 2 ijms-24-06233-t002:** Pearson correlation coefficients among root length (RL), root surface area (SA), average of root diameter (RD), total seedling dry weight (TDW) and total P content (PCont) for 561 maize lines. Correlation coefficients along the diagonal are between low- and high-P levels for each trait. Values below diagonal are the correlations between traits evaluated under low-P levels. Above diagonal numbers are the correlations between traits evaluated under high-P.

Traits	RL	SA	RD	TDW	PCont
RL	**0.46 ***	0.67 *****	−0.36 *****	0.54 *****	0.22 *****
SA	0.75 *****	**0.72 ***	0.38 *****	0.18 *****	−0.05 ^ns^
RD	−0.02 ^ns^	0.60 *****	**0.92 ***	−0.39 *****	−0.29 *****
TDW	0.34 *****	0.14 *****	−0.18 *****	**0.54 ***	0.70 *****
PCont	−0.14 *****	−0.27 *****	−0.27 *****	0.58 *****	**0.42 ***

* Coefficients significant at *p* < 0.01; ^ns^ non-significant at *p* < 0.01.

**Table 3 ijms-24-06233-t003:** Effects of contrasts between low- and high-P levels and the genetic variance for each trait within global, Embrapa and DTMA panels. The positive sign of the effect indicates that the average under high-P conditions was superior to low-P conditions, whereas the negative sign indicates a higher average under low-P conditions compared to high-P conditions. The Embrapa panel was composed of 365 lines and the DTMA panel, of 196 lines, in a total of 561 maize lines.

Trait	Effects of P within Panels			Genetic Variance ^a^			
	Global	Embrapa	DTMA	Embrapa		DTMA	
				High-P	Low-P	High-P	Low-P
Root Length (cm)	3.27 *	−6.37 *	16.79 **	1529.56 (160.84)	1368.54 (147.86)	2595.15 (353.17)	2700.39 (353.06)
Root Surface Area (cm^2^)	2.81 ***	−2.42 ***	10.11 ***	69.97 (7.30)	43.64 (5.11)	311.36 (44.05)	348.23 (47.04)
Root Diameter (mm)	0.07 ***	0.07 ***	0.11 ^ns^	0.003 (0.0004)	0.003 (0.0004)	0.02 (0.002)	0.01 (0.001)
Total Dry Weight (mg)	9.89 ***	8.29 ***	13.04 ^ns^	319.55 (34.42)	221.47 (23.23)	185.61 (24.27)	147.87 (21.92)
Total P Content (mg)	0.02 ***	0.02 ***	0.02 ^ns^	0.04 (0.005)	0.02 (0.004)	0.02 (0.002)	0.01 (0.003)

*, ** and *** significant at α=0.05, 0.01 and 0.001, respectively. ns: non-significant. ^a^ Standard errors are presented within brackets.

**Table 4 ijms-24-06233-t004:** Significant SNPs identified with mixed linear models (MLM) analyses for root morphology traits, total seedling dry weight and P content for low- and high-P conditions, based on a –log_10_(*p*-value) ≥ 6.07. The bin, position, alleles, minor allele frequency (MAF) and nearby candidate genes with a maximum distance of 1 kb are shown.

Trait	P level	SNP	Chr (bin)	Position (Mb)	−log_10_(*p*-Value)	Alleles	MAF	Predicted Gene
RD	high	S4_3751192	4.01	37.51	6.44	A/C	0.21	GRMZM2G037472
RL	low	S8_88600375	8.03	88.60	6.30	C/A	0.13	GRMZM2G044531
TDW	low	S6_34607369	6.01	34.61	6.37	T/C	0.25	NA
	low	S9_137746077	9.06	137.75	6.43	G/A	0.48	GRMZM2G378852
	high	S10_77284783	10.03	77.28	6.41	G/C	0.09	GRMZM2G110145
PCont	high	S8_21326857	8.03	21.33	6.34	G/A	0.22	NA
	high	S9_143192439	9.06	143.19	6.11	T/A	0.47	GRMZM2G104618

## Data Availability

The data presented in this study are available on request from the corresponding author. The data are not publicly available due to restrictions on protected maize lines.

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
