# Peer review of "Genome-Wide Association Study for Root Morphology and Phosphorus Acquisition Efficiency in Diverse Maize Panels"

_ijms, 2023, doi:10.3390/ijms24076233_

Round 1

Reviewer 1 Report

Phosphorus is an essential element for plant growth and development, which composes nucleic acids and biomembrane, and plays a critical role in the maintenance of life movements. Root traits are also ralely used in plant breeding and genetic study due to its difficulty in elvaluation. This work explores the root plasticity responses under low- and high-P concentrations with a diverse tropical maize panel and identifies candidate genes associated with root morphology and P acquisition trais, which could be exploited to improve effeiciency in maize breeding programs. Some minor revisions or questions were suggested below.

1.    In the analysis of principal component, DTMA lines were divided into two groups representing lines from Latin America and Africa, representing the geographical origin and pedigree information. I am wondering if the DTMA lines from Latin American or from Africa could be further divided into two heterotic groups of Flint and Dent, rather than clusterted in a group, although they were gathering from one place. And what caused this gathering.

2.    In the Figure 2 and 3, a global panel appears without introduction in the Material and shows different results with Embrapa panel and DTMA panel. And Why do the root traits show different changes under high and low-P treatments within three panels. Are there any intrinsic causes or not?

3.    Phosphorus utilizaiton efficiency and acquisition efficiency are both important in the process of P use efficiency, and in the work how to decide the traits investigated here belong to acquisition mechanism?considering there are both high and low P treatments.

4.    In the maize, ZmPTF1 is a known gene that improve phosphorus efficiency. And if it was identified in your association analysis. In bin 8.03, GRMZM2G044531 for AGC kinase and its allele need to be further verified in the maize.

Author Response

Reviewer 1: Comments and Suggestions for Authors

Phosphorus is an essential element for plant growth and development, which composes nucleic acids and biomembrane, and plays a critical role in the maintenance of life movements. Root traits are also rarely used in plant breeding and genetic study due to its difficulty in evaluation. This work explores the root plasticity responses under low- and high-P concentrations with a diverse tropical maize panel and identifies candidate genes associated with root morphology and P acquisition traits, which could be exploited to improve efficiency in maize breeding programs. Some minor revisions or questions were suggested below.

  1. In the analysis of principal component, DTMA lines were divided into two groups representing lines from Latin America and Africa, representing the geographical origin and pedigree information. I am wondering if the DTMA lines from Latin American or from Africa could be further divided into two heterotic groups of Flint and Dent, rather than clustered in a group, although they were gathering from one place. And what caused this gathering.

Answer: This is an interesting point to explore. However, unfortunately, the information about the heterotic groups of DTMA lines is unavailable. All available data of the DTMA is presented in the Supplementary Table S1, which are the country or the adaptation zone, according to Cairns et al. (2013). On the other hand, the Embrapa lines were clustered in three groups, representing the major heterotic groups Flint and Dent, and an intermediate C group. These clusters could be partially explained by their different origin and pedigree data, reflecting a wide genetic diverse of our maize panel.

  1. In the Figure 2 and 3, a global panel appears without introduction in the Material and shows different results with Embrapa panel and DTMA panel. And why do the root traits show different changes under high and low-P treatments within three panels. Are there any intrinsic causes or not?

Answer: Thanks for the question. The description of the global panel was inserted in the methods. As Embrapa and DTMA panels were genetically divergent in our principal component analysis, allele frequency changes between these panels are expected. Hence, it is reasonable to expect that genes controlling root morphology, biomass and P content could also be differentially modulated by P availability in each panel. One possible causes of differences between panels is selection. A recent publication supports the importance of selection during maize domestication and breeding on root-related genes, which contributed to a root system suitable for high-density planting (Ren et al., 2022). The changes of root morphology and P-efficiency related traits under contrasting P treatments within both maize panels were discussed in the item 3.2, which was also improved in the text based on your question.

  1. Phosphorus utilization efficiency and acquisition efficiency are both important in the process of P use efficiency, and in the work how to decide the traits investigated here belong to acquisition mechanism?considering there are both high and low P treatments.

Answer: The reviewer is correct. The traits evaluated in our manuscript were essentially related to P acquisition efficiency (PAE). PAE was more important compared to internal P utilization efficiency to explain P use efficiency under low- and high-P soils (Parentoni; De Souza Junior, 2008)... Additionally, root morphology traits have been described to be highly important for sorghum grain yield under low-P soil (Bernardino et al. 2019; 2021).

  1. In the maize, ZmPTF1 is a known gene that improve phosphorus efficiency. And if it was identified in your association analysis. In bin 8.03, GRMZM2G044531 for AGC kinase and its allele need to be further verified in the maize.

Answer: Thank you for the question. ZmPTF1 encodes a bHLH transcription factor that was shown to improve low phosphate tolerance in maize by regulating root growth (Li et al., 2011). This gene, GRMZM2G024530, is located at 11.56 Mb on chromosome 9, where no association signal was found in our study. Our GWAS study identified genomic regions and candidate genes as targets for further validation studies, including the AGC kinase, GRMZM2G044531. The SNP S8_88600375, located within this candidate gene was associated with root length under low-P [–log10(p-value) = 6.30]. The possible role of GRMZM2G044531 in P efficiency was discussed in lines 310-315.

References

Bernardino, K.C. et al. Association mapping and genomic selection for sorghum adaptation to tropical soils of Brazil in a sorghum multiparental random mating population. Theor. Appl. Genet. 2021, 134, 295–312.

Bernardino, K.C. et al. The genetic architecture of phosphorus efficiency in sorghum involves pleiotropic QTL for root morphology and grain yield under low phosphorus availability in the soil. BMC Plant Biol. 2019, 19, 1–15.

Cairns, J.E. et al. Identification of drought, heat, and combined drought and heat tolerant donors in maize. Crop Sci. 2013, 53, 1335–1346.

Li, X. et al. Overexpression of transcription factor ZmPTF1 improves low phosphate tolerance of maize by regulating carbon metabolism and root growth. Planta 2011, 233,1129–1143.

Parentoni, S.N.; De Souza Júnior, C.L. Phosphorus acquisition and internal utilization efficiency in tropical maize genotypes. Pesqui. Agropecu. Bras. 2008, 43, 893–901.

Ren, W., et al. Genome-wide dissection of changes in maize root system architecture during modern breeding. Nat. Plants 2022, 8, 1408–1422.

Reviewer 2 Report

Phosphorus is one of the most important elements necessary for the normal development of plants. Therefore, the importance of studying the influence of P on development, its participation in various cellular processes is undeniable. In the submitted manuscript, a large experiment was done with mathematical processing and the use of software. But it lacks clarity in the presentation of the experiment itself, how the initial data were selected.

Table 1 presents the results of the sum of 561 lines of two Embrapa and DMTA. But then the difference in morphological parameters between them is discussed.

The table shows mean, min and max values. But it is not clear how many lines are in these groups and how the groups change quantitatively depending on the P concentration.

In table 2, the designation of the total length of the root, etc. is not clear. Is this the sum of all the lengths of the roots of 561 lines at different concentrations of P? What is its fundamental significance?

The use of mathematical calculations greatly decorates the work, but it is desirable to describe in detail how the calculation of each parameter was carried out.

There are some minor remarks

-39 - phosphorus is included not only in nucleic acids and phospholipids, but also in proteins and secondary metabolites.

-48, 53 - references by one parenthesis

-178,185 - abbreviation PC is used to denote different concepts

-307 - rephrase the sentence

Author Response

Reviewer 2: Comments and Suggestions for Authors

1) Phosphorus is one of the most important elements necessary for the normal development of plants. Therefore, the importance of studying the influence of P on development, its participation in various cellular processes is undeniable. In the submitted manuscript, a large experiment was done with mathematical processing and the use of software. But it lacks clarity in the presentation of the experiment itself, how the initial data were selected.

Answer: Thanks for the question. The description of methodology to generate the initial phenotypic data, including the software processing, was detailed in the methods section (lines 387-402). The rationale behind our choice of root morphology traits to be used in the present study is now described (lines 402-405).

2) Table 1 presents the results of the sum of 561 lines of two Embrapa and DMTA. But then the difference in morphological parameters between them is discussed. The table shows mean, min and max values. But it is not clear how many lines are in these groups and how the groups change quantitatively depending on the P concentration.

Answer: Table 1 shows the description of the phenotypic data in the global panel. As the Embrapa and DTMA panels were found to be genetically divergent based on a principal component analysis, it can be expected that genes controlling root morphology, biomass and P content could also be differentially modulated by P availability in each panel. Thus, the phenotypic data of the Embrapa and DMTA panels were evaluated separately. The results of statistical analysis of contrasts between low- and high-P levels in each panel were presented in Table 3. Additionally, all data of individual lines in each panel was depicted in Figures 2 and 3, in order to clearly show the means and the dispersion of phenotypic data, depending on the P concentration. The number of lines in each panel was added to the legend of Table 3. The changes of the phenotypic traits under low- and high-P availability within the Embrapa and DMTA panels were discussed in sub-section 3.1.

3) In table 2, the designation of the total length of the root, etc. is not clear. Is this the sum of all the lengths of the roots of 561 lines at different concentrations of P? What is its fundamental significance?

Answer: Thank you for the observation. The designation of the root traits was standardized as root length, root surface area and root diameter in the text. These root traits measurements were clearly explained in the methods. What we called “total”, was the evaluation of the complete root system of a single plant, comprising the primary, seminal and initial adventitious roots. The phenotypic traits were expressed as BLUEs (best linear predicted estimates) of three replicates representing each of 561 lines at different P concentrations. Table 2 represents the correlation coefficients between all pairs of phenotypic traits assessed under low- (below diagonal) and high-P (above diagonal) concentrations. The significance of the correlation coefficients indicates the strong relationship of the root morphology traits with biomass accumulation and P efficiency in maize seedlings.

4) The use of mathematical calculations greatly decorates the work, but it is desirable to describe in detail how the calculation of each parameter was carried out.

Answer: Mathematical calculations and software processing are indispensable for processing the phenotypic data, including extracting quantifiable root morphology traits from root images, and to undertake GWAS. Several insertions and corrections were made, mainly in the methods, to clarify and to detail the calculation of root morphology traits. It is important to emphasize that the results generated in our manuscript were comparable with the data generated in different maize panel (384 temperate lines of AMES panel) and with a different methodology to evaluate root system morphology (ARIA) (Pace et al. BMC Genomics, 16: 47, 2015).

5) There are some minor remarks

-39 - phosphorus is included not only in nucleic acids and phospholipids, but also in proteins and secondary metabolites.

Answer: Information included in the text.

-48, 53 - references by one parenthesis

Answer: Citation corrected.

-178,185 - abbreviation PC is used to denote different concepts

Answer: Thanks for this observation. PC, designated as principal component, was used for two purposes in the manuscript, for diversity analysis and for type-I error correction in the association model. Although being used for different purposes, the concept behind the method is the same. Hence, the acronym PC was maintained in the association models incorporating population structure. Some clarifications about the use of principal component analysis were added in the text.

-307 - rephrase the sentence

Answer: Thanks for the suggestion, and the sentence was reworded. MAPKKK is a component of mitogen-activated protein kinase cascade involved on signal transduction to translate external stimuli into cellular responses. Several MAPKKKs were involved in signaling pathways in different maize organs and developmental stages [56].